



# Monthly-averaged maps of surface BRDF parameters in ten spectral bands for land and water masses

Philippe Blanc[1], Benoit Gschwind[1], Lionel Ménard[1], Lucien Wald[1]

[1]MINES ParisTech, PSL Research University, O.I.E. - Centre Observation, Impacts, Energy, 06904 Sophia Antipolis, France

*Correspondence to*: Philippe Blanc (philippe.blanc@mines-paristech.fr)

**Abstract.** The construction of worldwide maps of surface bidirectional reflectance distribution function (BRDF) parameters is presented. The original data stems from the National Aeronautics and Space Administration (NASA) of the USA which is making available maps of BRDF parameters that are derived from the Moderate Resolution Imaging Spectroradiometer (MODIS) instrument. The first parameter *fiso* describes the isotropic part of the BRDF while the two others *fvol* and *fgeo* describe the anisotropic part and are linked to the viewing and illuminating geometry. The original data has been averaged for each calendar month for the period 2004-2011 and a spatial completion of data was performed, especially in water-covered areas. The resulting complete maps are available in ten spectral bands: [459-479] nm, [545-565] nm, [620-670] nm, [400-700] nm, [841-876] nm, [1230-1250] nm, [1628-1652] nm, [2105-2155] nm, [250-5000] nm, [700-5000] nm, [250-5000] nm. The maps form a Global Earth Observation System of Systems (GEOSS) Data Collection of Open Resources for Everyone (Data-CORE) supporting the GEOSS Data Sharing Principles. They are referenced by the doi:10.23646/85d2cd5f-ccaa-482e-a4c9-b6e0c59d966c and available in NetCDF format under the Creative Commons license CC-BY.

## 1 Introduction

The reflectance of the surface is an important variable in many physical processes encountered in various domains such as meteorology, climate, or phenology. It depends upon the type of surface itself and its properties, the illuminating and viewing geometry, and the wavelength of the illuminating wave. The bidirectional reflectance distribution function (BRDF) describes how the reflectance varies with this geometry.

The BRDF determines how and to what extent a radiance originating from a given point on the sky vault will be reflected (Schaepman-Strub et al., 2006). The reflection of the downward solar radiation by terrestrial objects (ground, atmospheric constituents) is a key process in many aspects of remote sensing techniques and exploitation of data acquired by satellites; this underlines the importance of an accurate knowledge of the BRDF. The downwelling solar irradiance observed at ground level on horizontal surfaces is called global irradiance $E$ and is the sum of the direct and diffuse irradiances. The direct irradiance $B$ is received from the direction of the sun, including part of the circumsolar irradiance (Blanc et al., 2014). The diffuse irradiance $D$ originates from the rest of the sky vault. If they are integrated over the whole spectrum, these irradiances



are called total irradiances. Part of the diffuse irradiance is composed of photons that were once or more reflected by the ground and then by the atmosphere, cloudy or not, towards the receiving horizontal surface. Hence the diffuse and the global irradiances are sensitive to the BRDF. The surface albedo is defined as the ratio of the irradiance reflected by the surface to $E$. Because of the importance of the solar radiation in many domains, and of this dependence of the reflected light to the

BRDF, there is a strong need to know the BRDF, and further the surface albedo, at any time anywhere.

Though several attempts were made decades ago (see e.g. Moussu et al., 1989; Pinker and Razgaitis, 1983; Pinker and Ewing, 1986; Taylor and Stowe, 1984), such knowledge was not easily available worldwide before the advent of the Moderate Resolution Imaging Spectroradiometer (MODIS) instrument aboard the Terra and Aqua satellites of the National Aeronautics and Space Administration (NASA) of the USA (Schaaf et al., 2002; Strahler et al., 1999). Multi-angular

observations of any given location are obtained in several spectral bands and yield the BRDFs for these bands and a given period worldwide. Currently, the MODIS-derived BRDF is modelled with three parameters only as it will be seen later and these three BRDFs parameters are made available to anyone.

For decades, MINES ParisTech has been involved in the operational assessment of the downwelling solar irradiances at surface, $E$, $B$ and $D$. Irradiances may be obtained by the application of the Heliosat-2 method (Lefèvre et al., 2007; Rigollier

et al., 2004) onto images taken by geostationary satellites (Blanc et al., 2011; Diabaté et al., 1988). The surface albedo plays a major role in this assessment (Albarelo et al.; 2015; Demain et al., 2013; Espinar et al., 2009; Lefèvre et al., 2007). In the first versions of Heliosat, the ground was considered as Lambertian in nature, except for glitter conditions over the ocean (Wald and Monget, 1983), and the albedo was assessed as a statistical estimation of a minimal observed reflectance at each pixel during a given period. The limitations of this approach and the importance of an error on the surface albedo have been

underlined by several authors, such as Albarelo et al. (2015), Demain et al. (2013), Espinar et al. (2009) or Lefèvre et al. (2007). The MODIS-derived BRDF offers a unique opportunity to improve this operational assessment towards a better accuracy by taking into account the anisotropy of the reflection by the surface. In addition, it offers a first step towards a worldwide coverage.

One difficulty is that the global albedo products proposed by the NASA have missing values, notably due to clouds.

=Missing data implies more complexity in a completely automated chain as it imposes constant checks of the data availability and a strategy for dealing with these cases. Being involved in such automated processes, MINES ParisTech had to cope with the problem of missing data in the MODIS-derived data sets. To solve this issue, MINES ParisTech has created multi-annual averages of spatially gap-free maps of monthly means of BRDF parameters. These maps also have also the benefit of being a considerably smaller static BRDF wordwide database that is much easier to handle in an operational

context. The objective of the article is to describe how these maps were created and to share these data sets which form a Global Earth Observation System of Systems (GEOSS) Data Collection of Open Resources for Everyone (Data-CORE) supporting the GEOSS Data Sharing Principles.



## 2 Data and method

MODIS is an instrument aboard the Terra and Aqua satellites of the NASA. It acquires data in several spectral bands, from which among others, three parameters describing the surface BRDF are derived (Schaaf et al. 2002; Strahler et al., 1999) for each spectral band. The first parameter *fiso* describes the isotropic part of the BRDF while the two others *fvol* and *fgeo*

describe the anisotropic part. The directional hemispherical reflectance $\rho_{bs}$ is the reflectance of the surface under illumination from a single direction with no diffuse component and is consequently often termed black-sky albedo (Schaepman-Strub et al., 2006). On the contrary, the bihemispherical reflectance $\rho_{ws}$ is the reflectance of the surface under diffuse illumination only and is often termed white-sky albedo (Schaepman-Strub et al., 2006). For each spectral band, $\rho_{bs}$ and $\rho_{ws}$ are computed from *fiso*, *fvol* and *fgeo* using the following formulas, where $\theta_S$ is the solar zenithal angle:

$$\rho_{bs}(\theta_S) = fiso + 0.189184\ fvol - 1.377622\ fgeo \tag{1}$$

$$\rho_{ws}(\theta_S) = fiso + fvol\ (-0.007574 - 0.070987\ \theta_S^2 + 0.307588\ \theta_S^3) + fgeo\ (-1.284909 - 0.166314\ \theta_S^2 + 0.041840\ \theta_S^3) \tag{2}$$

Eventually, the surface albedo $\rho_g$ is given by:

$$\rho_g(\theta_S) = \rho_{ws} + [B / E(\rho_g)]\ (\rho_{bs} - \rho_{ws}) = \rho_{bs} + [D / E(\rho_g)]\ (\rho_{ws} - \rho_{bs}) \tag{3}$$

The US Geological Survey (USGS) web site proposes two combined (global) products MCD43C1 and MCD43C2 of the

three BRDF model parameters. They are derived from MODIS images and are 16-day composites produced every 8 days with a moving window of 16 days of acquisition, where the given date is that of the first day of the 16 days period. These level-3 products are projected onto a 0.05° grid in latitude/longitude, approximately 5.6 km at Equator.

The three parameters are available for ten spectral bands (Table 1). Seven of them are the original bands of the MODIS sensor while the NIR, shortwave and Vis bands have been created by assembling the original MODIS bands (Liang et al.,

20   1999).

| Band | Interval (nm) | Band | Interval (nm) |
|---|---|---|---|
| MODIS band 1 (Red) | 620-670 | MODIS band 2 | 841-876 |
| MODIS band 3 (Blue) | 459-479 | MODIS band 4 (Green) | 545-565 |
| MODIS band 5 | 1230-1250 | MODIS band 6 | 1628-1652 |
| MODIS band 7 | 2105-2155 | NIR* | 700-5000 |
| Shortwave* | 250-5000 | Vis* | 400-700 |

Table 1. List of bands and their spectral interval. * means that the band was defined in Liang et al. (1999)

Both products MCD43C1 and MCD43C2 are worldwide but restricted to land. Both exhibit gaps in data that are irregular in space and time. The MCD43C2 product is a snow-free version of MCD43C1. Both were downloaded for the period 2004-2011 and further analysed for the missing data. It was found that 26.2% of the pixels have at least one valid value per



calendar month in MCD43C1 and only 18.4% in MCD43C2. Approximately 3.3% of the pixels bearing a valid value differ in value between MCD43C1 and MCD43C2, i.e. 96.7% known pixels agree. Hence, the difference in values is small between both products. The major difference is that MCD43C1 has more known pixels.

When describing the operational model McClear for assessing the surface irradiance under cloud-free conditions, Lefèvre et

al. (2013) have reported that using multiannual monthly means instead of the product MCD43C2 did not change noticeably the accuracy of the retrieved irradiances and make operations more tractable. MINES ParisTech has built on this result in order to cope with the missing BRDF data, and has decided to use only multiannual means of monthly averages of the BRDF parameters computed over the period 2004-2011. The mean values of the three BRDF parameters from MCD43C1 product were computed for each of the twelve calendar months over the period 2004-2011 for each band. Each mean value was

allotted to the 15th of each month. From an operational point of view, the triplets $f=(fiso, fvol, fgeo)$ for the current day are computed by linear interpolation between the two nearest means modulo 12, each mean value being assigned to the 15th of the corresponding month.

The MODIS "Land Cover Type Climate Modelling Grid" product named MCD12C1 is also available in the Web site. It provides the dominant land cover types in the same geographical grid than MCD43C1. In addition, it also provides a land

cover type assessment, percent distribution, and quality control information. It includes an identification of pixels containing water and the proportion of water in these pixels. It has a different period than MCD43C1: 2001-2009 instead of 2004-2011. Let $W$ be the binary mask for water and $P$ be the proportion of water in these water pixels. For a given calendar month, let $W$ a binary water mask set to 1 if the pixel has been classified as "water" at least once in the month during 2001-2009 and is set to 0 otherwise. Let $P$ be the average proportion of the water during the given month ($P>0$ when $W=1$ and $P=0$ when $W=0$).

After multi-annual averages, all pixels where $W=0$ exhibit valid triplets $f$ for each of the twelve calendar months. A large amount of pixels that have contained water at least once ($W=1$, and called later water pixels) had missing data and required specific multi-stage data completion. It happens that several pixels ($W=1$, $P=1$) exhibit valid triplets $f$ in the MCD43C1, mostly at latitudes comprised between -45° and + 45°. Let $fW$ the typical triplet for water pixels defined as the mode of the set of valid triplets $f$ of pixels fully covered by water ($W=1$ and $P=1$).

The completion method was performed in several steps as follows. Firstly, $fW$ was allotted to each unknown pixel where $W=1$ and $P=1$. Then, if a water pixel exhibited a valid triplet $f$ and was not fully covered by water, i.e. $P<1$, $f$ was replaced by the following linear combination: $P fW + (1-P) f$. This was performed for the whole world. At this stage, many data was still missing. A rolling average of +/-1 month (modulo 12) was then applied to pixels exhibiting gaps in time in order to fill in these gaps. Then, each still unknown pixel is filled by the median value of valid triplets in a spatial squared window of 11

pixels in size, for each month. The two previous steps are repeated for missing pixels respectively with temporal windows of +/-2 months and with a squared window of 21 pixels in size per month. The very few pixels still unknown were located in the middle of the ocean, extreme southern part of the Antarctic Ocean, and Greenland. They were individually treated manually and were allotted the mean value of the triplets $f$ averaged in their neighbourhood. Eventually, all pixels were bearing a valid triplet.



A visual inspection of each of the 120 resulting maps was performed to control the quality and detect any anomaly. As an illustration, Figures 1 and 2 display the maps of *fiso* in the shortwave band for January and July. The colour table is the same for both maps to allow visual comparison.

## 3 The BRDF data set: description and how to access it

A series of 120 worldwide maps of the BRDF parameters *fiso*, *fvol* and *fgeo* have been computed for the ten spectral bands and for each calendar month. They are complete in space, i.e. each grid cell every 0.05° bears a valid triplet. The maps form a GEOSS Data-CORE. They are available in NetCDF format under the doi:10.23646/85d2cd5f-ccaa-482e-a4c9-b6e0c59d966c under a Creative Commons license (CC-BY).

To increase the awareness on this data set, an ISO 19139 metadata record has been created in the catalogue of the Energy GEO community at webservice-energy.org. GEO stands for Group on Earth Observation and is the international initiative organising the GEOSS. The use of the catalogue allows a wider dissemination towards the GEO community thanks to the its weekly harvesting by the GEO Discovery and Access Broker (DAB). Consequently the BRDF maps are available for search and discovery on the GEO Portal for the benefit of the community.


## 4 Conclusion

For its own needs, MINES ParisTech has created monthly-averaged maps of surface BRDF parameters in ten spectral bands for land and water masses that are spatially gap-free. These maps were originally created for the computation of the downwelling solar irradiance at surface in the shortwave spectral band and are in operational use since 2015 in the McClear

model (Lefèvre et al. 2013) and the Heliosat-4 method (Qu et al., 2017) within the Copernicus Atmosphere Monitoring Service (CAMS) with many applications (Lefèvre et al., 2014). No trouble in operation was noted and no drawback traceable back to these maps was reported by users. The maps in the various spectral bands could be used in other domains, such as UV for human health or PAR for agrometeorology. This is why MINES ParisTech has completed its effort for other spectral bands and is sharing the results.

## 5 Sources of data used

The MCD43C1, MCD43C2 and MCD12C1 products are freely available at https://lpdaac.usgs.gov/dataset_discovery/modis/modis_products_table.



**Author contribution**

P.B. and L.W. conceived the study; P.B. and B.G. conceived the methods and performed the calculations; B.G., L.M. and L.W. organized the data and metadata for dissemination; all authors contributed to the paper.

**Competing interests**

The authors declare no conflict of interest. The founding sponsors had no role in the design of the study; in the collection, analyses, or interpretation of data; in the writing of the manuscript, and in the decision to publish the results.

**Acknowledgements**

The research leading to these results has received funding from the European Union's Seventh Framework Programme (FP7/2007-2013) under Grant Agreement no. 218793 (MACC project) and no. 283576 (MACC-II project). This work could
not have been done without the efforts made by NASA and the MODIS Team.

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

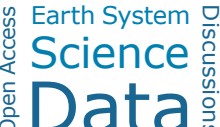



# Figures and captions

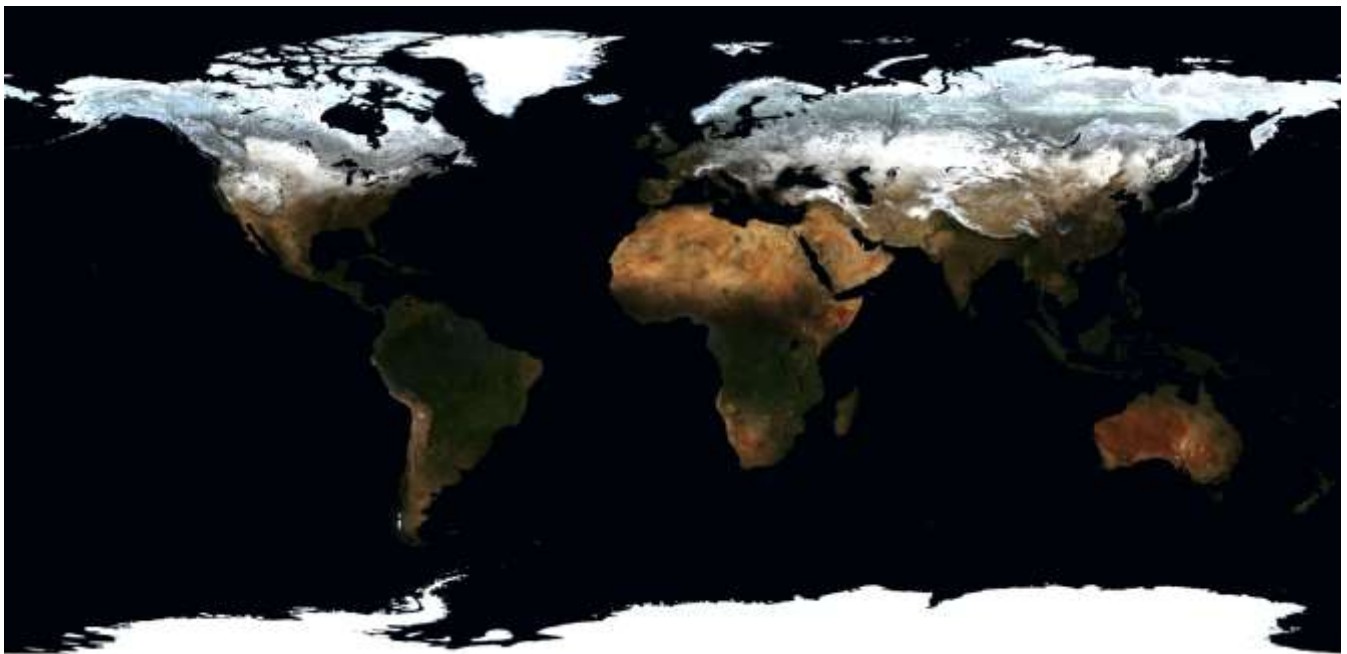

5    Figure 1. Composite RGB map of *fiso* in the shortwave band for January (Red: MODIS band 1, Green: MODIS band 4, Bleu: MODIS band 3, *cf.* Table 1)..

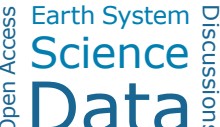

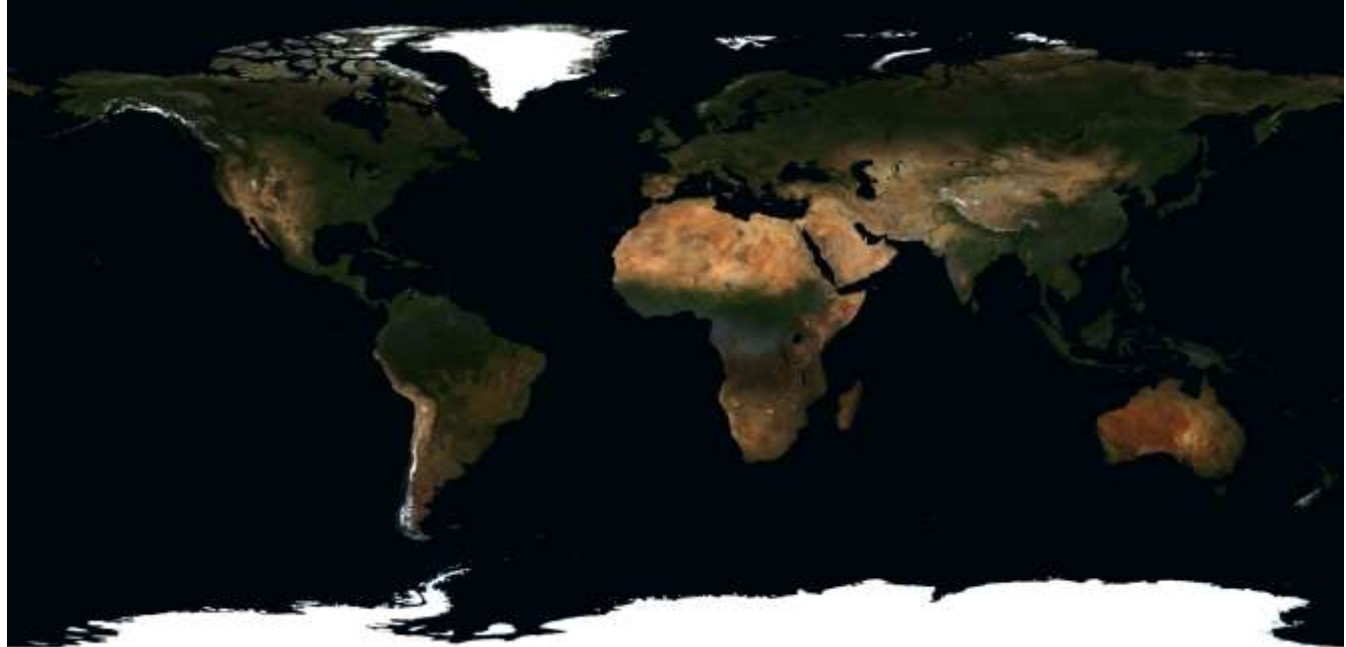

Figure 2. Composite RGB map of *fiso* in the shortwave band for July ((Red: MODIS band 1, Green: MODIS band 4, Bleu: MODIS band 3, *cf.* Table 1).