# Peer review of "Monthly-averaged maps of surface BRDF parameters in ten spectral bands for land and water masses"

_Earth System Science Data, 2017_

## Short Comment (SC1) · 9 Jan 2018

This looks like it will be a handy source of gap-filled climatological BRDF parameters for albedo calculations.

While reading this manuscript, I noticed that Equations 1 and 2 are swapped. Equation 1 is the calculation for white-sky albedo (pws), and Equation 2 is that for black-sky albedo (pbs). See e.g. Lucht et al (2000) at http://ieeexplore.ieee.org/stamp/stamp.jsp?tp=&arnumber=841980 for the original source of these equations.

White-sky albedo has no geometric dependence (so the theta_s can also be dropped from the left-hand side of that equation), while black-sky albedo does.

---

## Author Comment (AC1) · 10 Jan 2018

Authors wish to thank the reviewer for:

- his positive comment on the handiness of the proposed dataset of the global climatological gap free multispectral BRDF parameters

- his careful rereading that revealed a swap between the equation (1) and (2)

The revised paper will indeed correct this error:

$\rho_{ws} = fiso + 0.189184 fvol - 1.377622 fgeo$

and

$\rho_{bs}(\theta_S) = fiso + fvol(0.007574 - 0.070987\theta_S^2 + 0.307588\theta_S^3) + fgeo(-1.284909 - 0.166314\theta_S^2 + 0.041840\theta_S^3)$

---

## Referee Comment (RC1) · Anonymous Referee #1 · 17 Jan 2018

In this paper, the authors present their method that compiles 8 years of BRDF parameters from MODIS into a single climatological product. The procedure is extremely simple as it is a simple average with no analysis of the quality index of the input data, and no screening for outliers. The description of the original MODIS product makes it very clear that the quality indices should be used in the analysis. It is then rather worrisome that the authors do not even mention the existence of such quality index. Besides, there is no indication that the products build by their procedure contain information whether the pixel data has been directly obtained from the original MODIS data or whether it has been derived from spatial and/or temporal interpolation/extrapolation. The paper is short and most of it is an introduction on the usefulness of the product for the authors purpose and a description of the original MODIS data. There is no

discussion on the quality of the original data and how this quality is impacted by their procedure.

I am not convinced that the dataset that is presented in this paper shall be of interest to the community. Indeed, the generation procedure is rather simple so that many group may prefer to develop its own dataset, specifically tailored to its need. The lack of quality information in the proposed dataset makes this likely.

---

## Referee Comment (RC2) · Anonymous Referee #2 · 12 Feb 2018

Earth Syst. Sci. Data Discuss.,

https://doi.org/10.5194/essd-2017-141-RC2, 2018 © Author(s) 2018. This work is distributed under

the Creative Commons Attribution 4.0 License.

Received and published: 12 February 2018

This study presents a dataset of multiannual monthly average of BRDF parameters in the semi-empirical RossThick-LiSparse model adopted by the MODIS BRDF/Albedo/NBAR products. One goal is to provide spatially complete BRDF parameter maps over the globe. However, the following three critical flaws in the generation of this dataset hinder its value and appeal to the earth science community.

First, the monthly mean of BRDF parameters may smooth out the dynamics of the anisotropic characteristics over a lot of earth surface areas significantly. Particularly during some rapid changing phenomena, such as vegetation greening/browning periods, snow accumulation/melting periods, surface BRDF may change within calendar month. This monthly mean loses all the information on these surface anisotropy changes, while these changes are of critical interests and importance to many earth

science disciplines. Although the authors argument the rational of this monthly mean approach for gap filling in the global BRDF parameter maps by referencing Lefèvre et al. (2013), this reference is only interested in the accuracy and change of ground albedo for surface solar irradiance estimate. The ground albedo may change rapidly during the mentioned surface condition changes above. Even if the albedo of a surface may stay the same, the anisotropy may have changed indicating important surface structure changes. To publish this BRDF parameter datasets to the earth science community, the loss of anisotropy from monthly mean significantly nullify its intention as a BRDF product.

Second, the authors chose the version 5 of MODIS BRDF datasets as their input. Looking at the LP DAAC website, the newer version, version 6 of MODIS BRDF products seem available much earlier than the submission of this manuscript. By any means, the newer version of MODIS product should be used as input for the current publication. Also an important notice to be emphasized is that MCD43 V6 products provide MCD43D, the 30 arc-second CMG products of the globe, besides the MCD43C, the 0.05 degree CMG. In the user guide of the MCD43 product, I found that it states this MCD43D is direct retrieval from sensor observations (https://www.umb.edu/spectralmass/terra\_aqua\_modis/v006/mcd43d\_cmg\_30\_arc\_second\_products) while MCD43C is just average based on the 30 arc-second retrievals (https://www.umb.edu/spectralmass/terra\_aqua\_modis/v006/mcd43c1\_cmg\_brdf\_albedo\_model\_parameters\_product). This means that the MCD43C is already an average rather than retrieval of BRDF using the RossThick-LiSparse model. The quality of MCD43C is not as high as the MCD43D. To generate spatially complete product, an input of MCD43D is more appropriate.

Third, the MODIS BRDF/Albedo/NBAR product team has already provided a spatially complete CMG product at 30 arc-second gridding size over the globe (https://www.umb.edu/spectralmass/terra\_aqua\_modis/v006/mcd43gf\_cmg\_gap\_filled\_snow\_free\_products); and on LP DAAC (https://lpdaac.usgs.gov/dataset\_discovery/modis/modis\_products\_table).
Lastly, I couldn't find the data using the doi provided in the manuscript. And there is no web link to the data. After I searched on google, I found this page http://www.oie.mines-paristech.fr/Valorisation/Outils/AlbedoSol/. And downloaded the HDF5 file of BRDF parameters there. However, the single HDF5 file only provides BRDF parameters at the shortwave band. Also there is no metadata in the HDF5 file explaining the dataset at all. No quality data there either.

---

## Author Comment (AC2) · 21 May 2018

Earth Syst. Sci. Data Discuss.,

https://doi.org/10.5194/essd-2017-141-AC2, 2018 © Author(s) 2018. This work is distributed under

the Creative Commons Attribution 4.0 License.

philippe.blanc@mines-paristech.fr

Received and published: 21 May 2018

First point: indeed, compiling multiannual monthly averages from time series of BRDF parameters maps removes the intra-monthly and inter-annual variabilities. Nevertheless, authors believe that having a "static"gap-filled BRDF parameters for the 10 spectral bands of MODIS may be as handy for some Earth science application such as the Heliosat-4 and McClear method. On the occasion of the conference EGU 2018, a Monte-Carlo based comparison has been done between the MCD43C1 time series and the Here is the main conclusion: "[The database with gap-filled multiannual monthly averages of BRDF parameters] is in good agreement with MCD43C1 (Fig. 2), except some specific cases: most of deviations observed come from regions with high inter-annual and intra-monthly variabilities of snow cover: - Northern mid-latitudes during winter - Arctic regions during spring and fall - Himalaya, Southern Andes, Alps For these regions at these specific periods, we also observe a significant negative bias"

In order to provide the users with information about the variability lost during the monthly averaging process, an additional field has been added with the corresponding multiannual monthly standard deviation of the BRDF parameters.

Second point: This work has been done with the database MCD43C1 v5 and the primary intention was to serve our own purpose, especially in assessing the solar radiation available at ground. We have been approached by other potential users in other communities and have decided to share our existing database for the common good. The NASA has released in between the version 6 of MCD43C1. The same approach we used for MCD43C1 v5 could be applied and, in that case, a new version of our monthly average database.
Third point: following the link https://www.umb.edu/spectralmass/terra\_aqua\_modis/v006/mcd43gf\_cmg\_gap\_filled\_snow\_ it is said that the gap free products MCD43GF will be produced from V006 MCD43D. The one compiled from V005 is indeed available at ftp://rsftp.eeos.umb.edu/data02/Gapfilled/ at 30 arcsec, every 8 days, from 2003 to 2015.This product is very interesting, but authors believes that having smaller and easy to access database of multiannual monthly averages of BRDF parameters maybe useful for some users in the community of Earth Science, just like it is useful for our own purpose.

Last point: at the time of the reviewer's check, our servers seemed to be down. Authors will double check the availability of the database along with the metadata. Comments about quality information have been addressed in the answer for the Anonymous Referee #1.

---

## Author Comment (AC3) · 21 May 2018

Authors recognize that the provided quality flags (BRDF_Quality) in MCD43C1 should be used in the computation of the multiannual monthly means of the BRDF parameters. For that purpose, authors have recompiled to database, taking into account the different level of quality provided by the BRDF_Quality flags. A specific incremental average algorithm (described pages 4 and 5) has been applied to compute means (and corresponding standard deviations) with the BRDF parameters with the best available quality (owing to the quality flags) and statistically sufficient elements. Out of this computation, in addition to the computation of averages, a quality table is provided for each pixel at each calendar month with the number of elements per class of BRDF_Quality flags that have been used for the computation.